# Pl@ntNet-300K: a new plant image dataset for the evaluation of set-valued classifiers

Camille Garcin[*1], Alexis Joly[†2], Pierre Bonnet[‡3], Antoine Affouard [2,3], Jean-Christophe Lombardo [2], Mathias Chouet[2,3], Maximilien Servajean[§4], and Joseph Salmon[¶ 5]

[1]IMAG, Univ Montpellier, Inria, CNRS, Montpellier, France
[2]Inria, LIRMM, Univ Montpellier, CNRS, Montpellier, France
[3]CIRAD, AMAP
[4]LIRMM, AMIS, UPVM, Univ Montpellier, CNRS, Montpellier
[5]IMAG, Univ Montpellier, CNRS, Montpellier, France

## Abstract

This paper presents a novel image dataset with high intrinsic ambiguity specifically built for evaluating and comparing set-valued classifiers. This dataset, built from the database of Pl@ntnet citizen observatory, consists of 306,146 images covering 1,081 species. We highlight two particular features of the dataset, inherent to the way the images are acquired and to the intrinsic diversity of plants morphology: i) The dataset has a strong class imbalance, meaning that a few species account for most of the images. ii) Many species are visually similar, making identification difficult even for the expert eye. These two characteristics make the present dataset well suited for the evaluation of set-valued classification methods and algorithms. Therefore, we recommend two set-valued evaluation metrics associated with the dataset (top-$k$ and average-$k$) and we provide the results of a baseline approach based on a deep neural network trained with the cross-entropy loss.

## 1 Introduction

The difficulty in classifying images comes from two main types of uncertainty [1]: i) the aleatoric uncertainty that arises from the intrinsic randomness of the underlying process, which is considered irreducible, and ii) the epistemic uncertainty that is caused by a lack of knowledge and is considered to be reducible with additional training data. In modern real-world applications, these two types of uncertainty are particularly difficult handle. The large number of classes tends to increase the class overlap (and thus the aleatoric uncertainty), and, on the other hand, the long tail distribution makes it difficult to learn the less populated classes (and thus increase the epistemic uncertainty). The presence of these two uncertainties is a central motivation for the use of set-valued classifiers, *i.e.,* classifiers returning a set of candidate classes for an image. A survey of existing methods for building set valued classifiers can be found in [2]. Although there are several datasets in the literature that have visually similar classes [3, 4, 5, 6], most of them do not aim to retain both the epistemic and the aleatoric ambiguity present in real world data.

---

[*]camille.garcin@inria.fr
[†]alexis.joly@inria.fr
[‡]pierre.bonnet@cirad.fr
[§]servajean@lirmm.fr
[¶]joseph.salmon@umontpellier.fr

Submitted to the 35th Conference on Neural Information Processing Systems (NeurIPS 2021) Track on Datasets and Benchmarks. Do not distribute.

In this paper, we propose a dataset designed to remain representative of real-world ambiguity, making it well suited for the evaluation of set-valued classification methods. This dataset is extracted from real world images collected by Pl@ntNet [7], a large-scale citizen observatory dedicated to the collection of plant occurrences data through image-based plant identification. The key feature of Pl@ntNet is a mobile application that allows citizen scientists to submit a picture of a plant to get a list of the most likely species for that picture. The application is used by more than 10 millions users in about 170 countries and is one of the main data publishers of GBIF [8], an international platform funded by the governments of many countries around the world to provide free and open access to biodiversity data. Another key feature of Pl@ntNet is that the training set used to train the classifier is collaboratively enriched and revised. Nowadays, Pl@ntNet covers over 35K species illustrated by nearly 12 million validated images.

The entire Pl@ntNet database would be an ideal candidate for the evaluation of set-valued classification methods, but it is far too large to allow for widespread use by the machine learning community. Thus, the dataset presented in this paper is constructed by retaining only a subset of the genera in the entire Pl@ntNet database (sampled uniformly at random) while retaining the species that belong to these genera. Retaining all species in a genus is intended to preserve the large amount of ambiguity present in the original database, as species in the same genus are likely to share common visual features.

The rest of the paper is organized as follows: we first introduce the set-valued classification framework in Section 2, focusing on two special cases: top-$k$ classification and average-$k$ classification. In Section 3, we describe the construction of the dataset, and show that it contains a large amount of ambiguity. Next, we describe in Section 4 the metrics of interest for the Pl@ntNet-300k dataset and propose benchmark results for these metrics, obtained by training several state-of-the art neural networks architectures. In Section 5, we compare Pl@ntNet-300K to several existing datasets. Finally we provide the link to the dataset in Section 6 before concluding.

## 2  Set-valued classification

We adopt the classical statistical set up of multi-class classification. Random couples of image and label $(X, Y) \in \mathcal{X} \times \{1, \ldots, d\}$ are assumed to be generated by an unknown joint distribution $\mathbb{P}$. The integer $d$ will denote the number of classes, and $[d]$ will refer to $\{1, \ldots, d\}$. In the following, $k \in [d]$. A set-valued classifier $\Gamma$ is a function mapping the feature space $\mathcal{X}$ to the set of all subsets of $[d]$, $2^{[d]}$, $\Gamma : \mathcal{X} \to 2^{[d]}$. Our goal is to build a classifier with low risk $\mathbb{P}(Y \notin \Gamma(X))$. However it is not desirable to simply minimize the risk: a set-valued classifier that always returns all of the classes achieves zero risk, but is useless. A set-valued classifier is useful if it returns only the most probable classes given a query image. Therefore a quantity of interest will be $|\Gamma(x)|$, the number of classes returned by the classifier $\Gamma$, given an image $x \in \mathcal{X}$.

In this section we will examine two optimization methods that lead to different set-valued classifiers. Both of them aim to minimize the risk, but they differ in the way they constrain the set cardinality: either pointwise or on average.

For $x \in \mathcal{X}$, we define $p_l(x) = \mathbb{P}(Y = l | X = x)$, and estimators of these quantities will be denoted by $\hat{p}_l(x)$. Finally, for $x \in \mathcal{X}$, we define the $\mathrm{top}_\mathrm{p}$ operator as: $\mathrm{top}_\mathrm{p}(x, k) = \{p_{\sigma_x(1)}(x), p_{\sigma_x(2)}(x), \ldots, p_{\sigma_x(k)}(x)\}$, where $\sigma_x : [d] \to [d]$ orders $\{p_1(x), \ldots, p_d(x)\}$ in decreasing order: $p_{\sigma_x(1)}(x) \geq p_{\sigma_x(2)}(x) \geq \cdots \geq p_{\sigma_x(d)}(x)$.

The most straightforward constraint is to require the number of classes returned to be less than $k$ for every input. This results in the following optimization problem :

$$\Gamma^*_\text{top-k} \in \arg\min_\Gamma \mathbb{P}(Y \notin \Gamma(X))$$
$$\text{s.t. } |\Gamma(x)| \leq k, \ \forall x \in \mathcal{X} \ . \tag{1}$$

The estimation of the risk, given this point-wise constraint, is known as top-$k$ error [9].

There is a closed form solution to Problem (1) [10] which is:

$$\Gamma^*_\text{top-k}(x) = \mathrm{top}_\mathrm{p}(x, k) \ . \tag{2}$$

This is the Bayes classifier. However this is not practical since we do not know $\mathbb{P}$. The plug-in estimator $\hat{\Gamma}_\text{top-k}$ naturally follows from (2): $\hat{\Gamma}_\text{top-k} = \mathrm{top}_{\hat{\mathrm{p}}}(x, k)$. While the top-$k$ accuracy is often

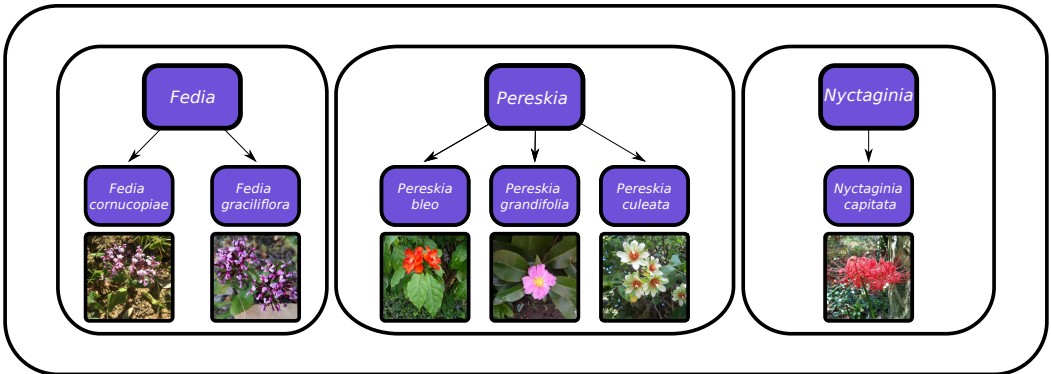

Figure 1: Genus taxonomy : we display three genus present in the proposed dataset : *Fedia*, *Pereskia* and *Nyctaginia*, which contain respectively two, three and one species.

reported in benchmarks, only a few works aim at directly optimizing that metric [9, 11, 10, 12]. An obvious limitation of top-$k$ classification is that $k$ classes are returned for every example, regardless of the difficulty of classifying that example. Average-$k$ classification allows for more flexibility. In that setting, the constraint on the size of the predicted set is more flexible and must be satisfied only on average. The optimization problem then becomes:

$$\Gamma^*_{\text{average-k}} \in \arg\min_{\Gamma} \mathbb{P}(Y \notin \Gamma(X))$$
$$\text{s.t. } \mathbb{E}_X |\Gamma(X)| \leq k. \tag{3}$$

The closed form solution is derived in [2] :

$$\Gamma^*_{\text{average-k}}(x) = \{l \in [d], p_l(x) \geq G^{-1}(k)\} \ , \tag{4}$$

where $G$ is defined as follows:

$$\forall t \in [0, 1], G(t) = \sum_{l=1}^{d} \mathbb{P}(p_l(X) \geq t) \ , \tag{5}$$

and $G^{-1}$ refers to the generalized inverse of $G$:

$$G^{-1}(u) = \inf\{t \in \mathbb{R}, G(t) \leq u\} \ . \tag{6}$$

Note that if we define the classifier $\Gamma_t$ by: $\forall x \in \mathcal{X}, \Gamma_t(x) = \{l \in [d], p_l(x) \geq t\}$, then $G(t)$ is the average number of classes returned by $\Gamma_t$: $G(t) = \mathbb{E}_X |\Gamma_t(X)|$. From (4) we see that the Bayes classifier corresponds to a thresholding operation. All classes having a conditional probability greater than $G^{-1}(k)$ are returned, where the threshold is chosen so that $k$ classes are returned on average. To compute the plug-in estimator, we first have to estimate $G$ with an unlabeled dataset $x'_1, x'_2, \ldots, x'_N : \hat{G}(t) = \frac{1}{N} \sum_{i=1}^{N} \sum_{l=1}^{d} \mathbb{1}[\hat{p}_l(x'_i) \geq t]$. The definition of the plug-in estimator then follows: $\hat{\Gamma}_{\text{average-k}}(x) = \{l \in [d], \hat{p}_l(x) \geq \hat{G}^{-1}(k)\}$, where $\hat{G}^{-1}$ refers to the generalized inverse of $\hat{G}$.

## 3 Dataset

### 3.1 Construction

In the biological classification of plants, species are organized into genera. Each genus contains several species, and the different genera do not overlap. A schema is proposed in Figure 1.

Instead of retaining randomly selected species or images from the entire Pl@ntNet dataset, we choose to retain randomly selected genera and keep all species belonging to these genera. This

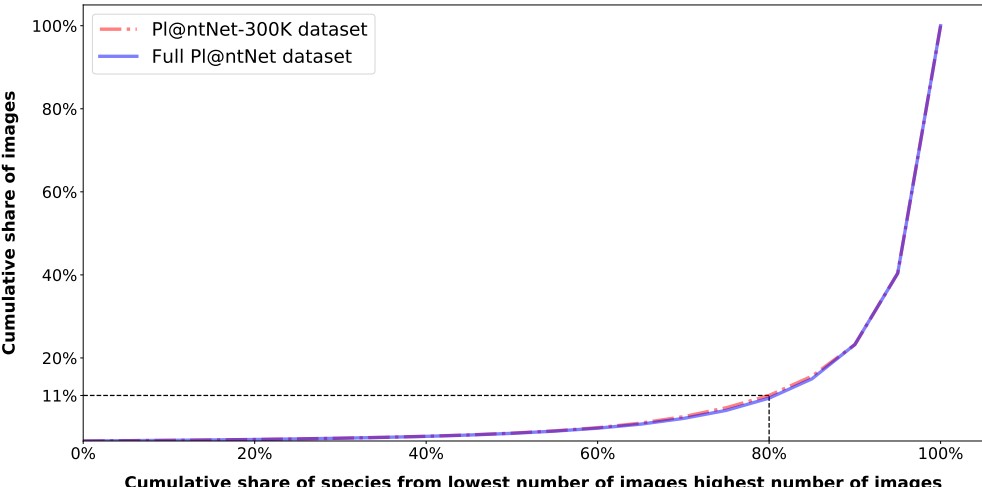

Figure 2: Lorenz curves of the Pl@ntNet database and the proposed dataset. Note that for fair comparison, we discard species with less than 4 images in the Pl@ntNet database.

choice aims to preserve the large amount of ambiguity present in the original database, as species belonging to the same genus tend to share visual features. The dataset presented in this paper is constructed by retaining only 10% of the genera of the whole Pl@ntNet database (sampled uniformly at random).

We then retain only species with more than 4 images, resulting in a total of 303 genera and $d = 1{,}081$ species, representing $n_{tot} = n_{train} + n_{val} + n_{test} = 306{,}146$ color images. At this point we have pairs of image and label $(x_i, y_i)_{i=1,\ldots,n_{tot}}$ with $y_i \in [d]$. The average image size is $(570, 570, 3)$, ranging from $(180, 180, 3)$ to $(900, 900, 3)$.

The images are divided into a training set, a validation set and a test set. The division is performed at the species level due to the long tail distribution. For each species, 80% of the images are placed in the training set ($n_{train} = 243{,}916$), 10% in the validation set ($n_{val} = 31{,}118$), and 10% in the test set ($n_{test} = 31{,}112$), with at least one image of each species in each set.

## 3.2 Epistemic uncertainty

In our case, epistemic uncertainty refers mainly to the lack of data necessary to properly estimate the conditional probabilities.

In Pl@ntnet, the most common species are readily available to users and thus represent a large fraction of the images, while the rarest species are more difficult to find and therefore more rare in the database.

The construction of the dataset described above preserves the class imbalance. To show this, we plot the Lorenz curves [13] of the entire Pl@ntNet dataset and of the Pl@ntNet-300K images dataset in Figure 2. In the proposed dataset, 80% of the species (the ones with the lowest number of images) account for only 11% of the total number of images. This poses a challenge when training learning models, since for many classes the model only has a handful of images to train on, making identification difficult for these species.

In addition to the long-tail distribution issue, epistemic uncertainty also arises from the high intra-species variability. Plants may take on different appearances depending on the season (flowering time). Furthermore, a user of the application may photograph only a part of the plant (for instance, the trunk and not the leaves). As a last example, flowers belonging to the same species can have different colors. These cases are illustrated in Figure 3 and contribute to a high intra-class variability which, combined with the long tailed distribution, makes it more challenging to model the species.

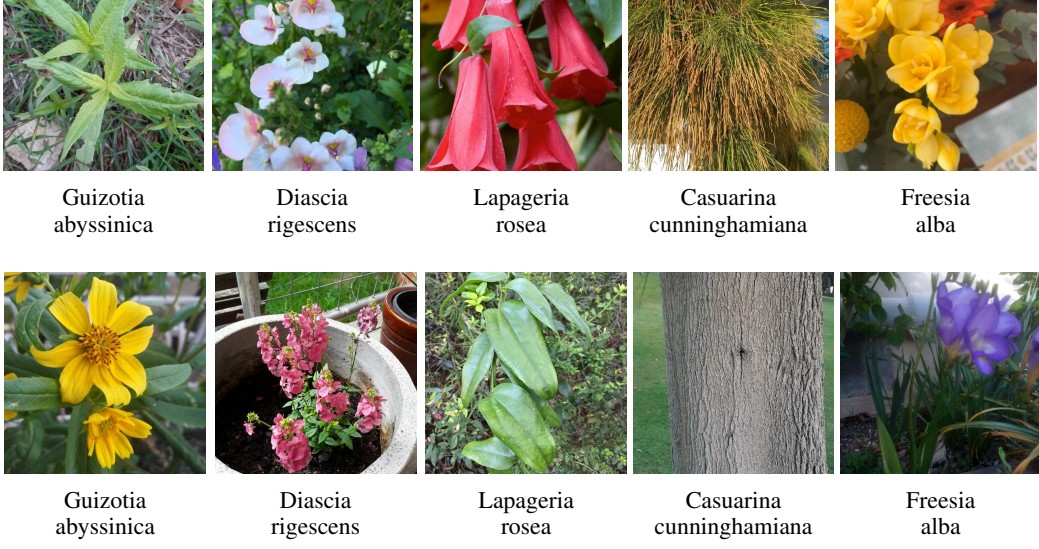

| Guizotia abyssinica | Diascia rigescens | Lapageria rosea | Casuarina cunninghamiana | Freesia alba |

Figure 3: Examples of visually dissimilar images belonging to the same class

## 3.3 Aleatoric uncertainty

In our case, the source of aleatoric uncertainty mostly resides in the limited information we are given to make a decision (*i.e.,* assign a class to a plant). Some species, especially those belonging to the same genus, can be visually very similar. For example, consider the case where two species produce the same flowers but different leaves, typically because they have evolved differently from the same parent species. If a person photographs only the flower of a specimen of one of the two species, then it will be impossible, even for an expert, to know whether it is one or the other species. The discriminating information is not present in the image.

The combination of this irreducible ambiguity with images of non-optimal quality (non-adapted close-up, low-light conditions, etc.) results in pairs of images that belong to different species but are difficult or even impossible to distinguish, see Figure 4 for illustration. In the figure, we show the ambiguity between pairs of species, but we could find similar examples involving a larger number of species. Thus, even an expert botanist might fail to assign a label to such pictures with certainty. This is embodied by $p_l(x)$ : given an image, multiple classes are possible.

## 4 Evaluation

### 4.1 Metric

To evaluate set valued predictors on Pl@ntNet-300k, we will examine two main different metrics: top-$k$ accuracy (as a baseline) and average-$k$ accuracy. Top-$k$ accuracy [11] is a widely used metric which is computed on the test set as follows :

$$top\text{-}k\ accuracy = \frac{1}{n_{test}} \sum_{(x_i,y_i)\in\text{test set}} \mathbb{1}[y_i \in \hat{\Gamma}_{\text{top-k}}(x_i)],\ \text{s.t.}\ |\hat{\Gamma}_{\text{top-k}}(x_i)| = k\ , \tag{7}$$

where $\hat{\Gamma}_{\text{top-k}}$ is a set-valued classifier built with the training data.

Average-$k$ accuracy [14] is a metric which is evaluated as follows :

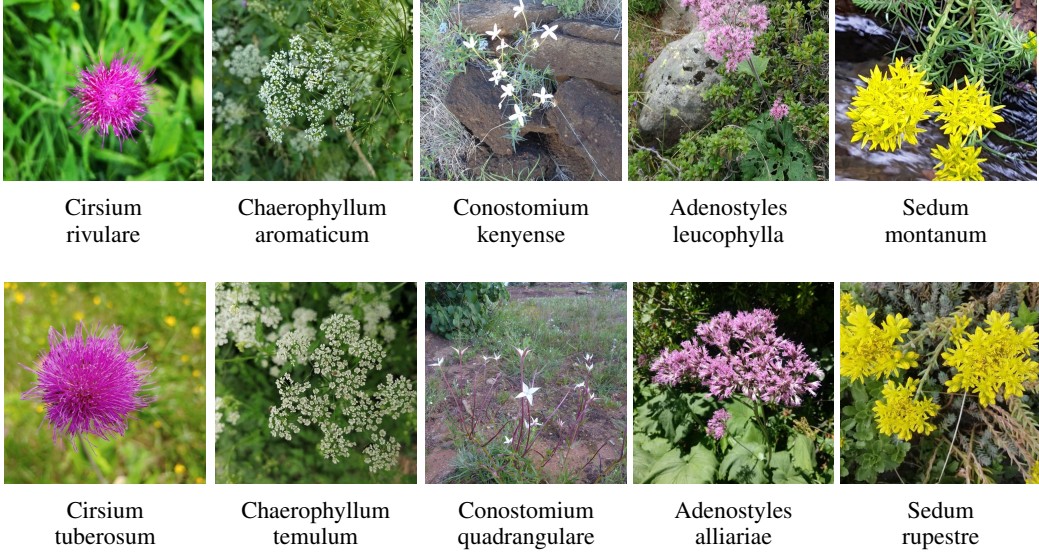

| Cirsium rivulare | Chaerophyllum aromaticum | Conostomium kenyense | Adenostyles leucophylla | Sedum montanum |
| Cirsium tuberosum | Chaerophyllum temulum | Conostomium quadrangulare | Adenostyles alliariae | Sedum rupestre |

Figure 4: Examples of visually similar images belonging to two different classes

$$\textit{average-k accuracy} = \frac{1}{n_{test}} \sum_{(x_i, y_i) \in \text{test set}} \mathbb{1}[y_i \in \hat{\Gamma}_{\text{average-k}}(x_i)] \text{ s.t. } \frac{1}{n_{\text{test}}} \sum_{x_i} |\hat{\Gamma}_{\text{average-k}}(x_i)| \leq k \ ,$$
(8)

149 where $\hat{\Gamma}_{\text{average-k}}$ is a set-valued classifier built with the training data.

150 The most straightforward to derive both classifiers is to first obtain an estimate of the conditional
151 probabilities $\hat{p}_l(x)$ and then derive the plug-in classifiers, as explained in Section 2. Our hope is
152 for the Pl@ntnet-300k dataset to encourage novel ways to derive the set-valued classifiers $\hat{\Gamma}_{\text{top-k}}$ and
153 $\hat{\Gamma}_{\text{average-k}}$ to optimize respectively the top-$k$ accuracy and the average-$k$ accuracy. Notice that a few
154 works already propose methods to optimize the top-$k$ accuracy [9, 11, 10, 12].

## 4.2 Baseline

156 In this section we provide a baseline evaluation of the plug-in classifiers. We train several state-of-the
157 art deep neural networks with the cross-entropy loss: ResNet-50 [15], DenseNet-121, DenseNet-169
158 [16], InceptionResNet-v2 [17] and MobileNetV2 [18].

159 First a pre-processing step is performed on the original images as follows: we extract the largest
160 centered square in the image and resize it to $299 \times 299$. No data augmentation is used. The model
161 are optimized for 70 epochs with SGD with a learning rate of $1.10^{-2}$, a momentum of 0.9 with the
162 Nesterov acceleration [19]. The learning rate is divided by 10 at epoch 40, 50 and 60. We use a
163 batch size is 64 for all models except for InceptionResNet-v2 and DenseNet-169 which are trained
164 with a batch size of 32. The criteria for early stopping is the accuracy on the validation set. For
165 the plug-in classifier $\hat{\Gamma}_{\text{average-k, plug-in}}$, we compute the threshold $\lambda_{val}$ on the validation set and use
166 that same threshold to compute the average-$k$ accuracy on the test set. All results in this section are
167 reported on the test set and are the result of an average over four different seeds.

168 We report accuracy, top-$k$ accuracy and average-$k$ accuracy in Table 1.

169 We also compute top-$k$ accuracy and average-$k$ accuracy for each class and report the average over
170 classes in Table 3. Table 2 illustrates the discrepancy between the accuracy (69.8% for ResNet-50)
171 and the average (over classes) of class accuracies (25.7% for ResNet-50), each class accuracy being
172 computed as the number of correctly classified examples in the class divided by the total number of
173 examples in the class. The difference can be explained by the long tail distribution. The model easily
174 identifies images from the most populated classes, which account for most images in the dataset, as

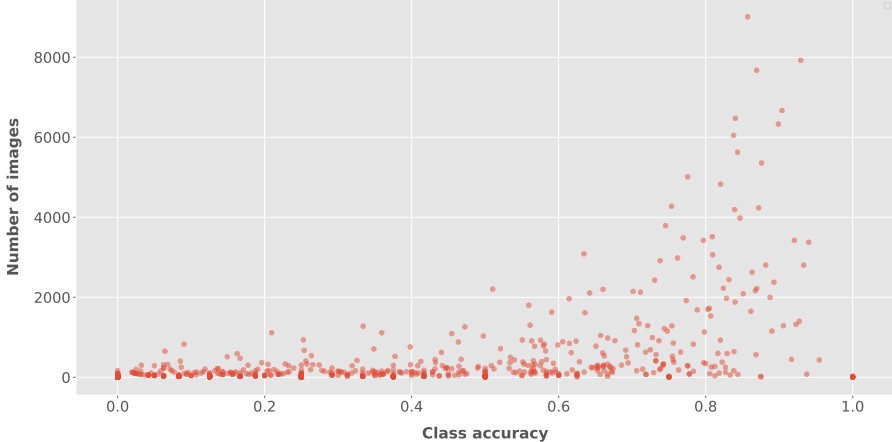

Figure 5: Relation between class accuracy and number of images in the class (evaluated on the test set with a ResNet-50).

|  | top-1 | avg-1 | top-3 | avg-3 | top-5 | avg-5 | top-10 | avg-10 |
|---|---|---|---|---|---|---|---|---|
| ResNet-50 | 69.8 | 70.3 | 84.2 | 84.1 | 88.2 | 87.2 | 91.9 | 90.7 |
| MobileNetV2 | 72.1 | 72.9 | 87.2 | 89.0 | 91.1 | 92.1 | 94.2 | 95.1 |
| DenseNet-121 | 75.7 | 76.1 | 89.5 | 90.1 | 92.7 | 92.7 | 95.4 | 95.2 |
| DenseNet-169 | 75.7 | 76.2 | 89.5 | 89.9 | 92.8 | 92.5 | 95.4 | 94.9 |
| InceptionResNet-v2 | 76.8 | 77.2 | 89.8 | 89.7 | 92.9 | 91.9 | 95.4 | 94.3 |

Table 1: Top-$k$ and Average-$k$ accuracy on the test set for different values of $k$ and various neural network architectures (evaluated on the test).

seen in Figure 2. When the class accuracies are averaged (over classes), species with few images (which are a majority, see Figure 2) significantly degrade the overall result. To support this claim, we show the correlation between class accuracy and number of images in Figure 5.

These results illustrate the difficulty of the Pl@ntNet-300k dataset. The accuracy (69.8% with a ResNet-50 [15]) is significantly lower than that of ImageNet (79.3% for a ResNet-50), which can be explained by the multiple sources of uncertainty described in Section 3. The main interest of these results is to show that the basic plug-in predictor obtained with the cross-entropy loss does not systematically give an average-k accuracy significantly better than the top-k accuracy (sometimes it is even worse, see Table 1), as one would expect. Thus, we believe there is a need for new average-$k$ prediction methods for improved performance.

## 5   Related work

Fined-Grained Visual Categorization (FGVC) is about discriminating visually similar classes. In order to better learn fine-grained classes, several approaches have been proposed by the FGVC community, including multi-stage metric learning [20], high order feature interaction [21, 22], and

|  | Accuracy | Average of class accuracies |
|---|---|---|
| ResNet-50 | 69.8 | 25.7 |
| MobileNetV2 | 72.1 | 28.1 |
| DenseNet-121 | 75.7 | 32.5 |
| DenseNet-169 | 75.7 | 32.5 |
| InceptionResNet-v2 | 76.8 | 32.4 |

Table 2: Accuracy and Average of class accuracies (evaluated on the test set).

|  | top-1 | avg-1 | top-3 | avg-3 | top-5 | avg-5 | top-10 | avg-10 |
|---|---|---|---|---|---|---|---|---|
| ResNet-50 | 25.7 | 25.9 | 45.5 | 50.7 | 55.8 | 59.3 | 67.9 | 69.8 |
| MobileNetV2 | 28.1 | 28.0 | 49.4 | 56.5 | 58.7 | 66.9 | 71.0 | 78.0 |
| DenseNet-121 | 32.5 | 32.3 | 55.0 | 61.8 | 65.0 | 70.6 | 75.9 | 79.6 |
| DenseNet-169 | 32.5 | 32.5 | 55.2 | 62.5 | 65.3 | 71.1 | 76.2 | 79.7 |
| InceptionResNet-v2 | 32.4 | 32.2 | 54.6 | 61.5 | 64.7 | 69.2 | 76.0 | 78.8 |

Table 3: Average (over classes) of top-k class accuracies and Average (over classes) of average-k class accuracies for different values of $k$ (evaluated on the test set).

| | Human-in-the-loop labeling | Long tail distribution | Intra-class variability | Focused domain | Ambiguity preserving sampling |
|---|---|---|---|---|---|
| CUB200 | x | x | x | ✓ | x |
| Oxford flower dataset | x | x | ✓ | ✓ | x |
| Aircraft dataset | ✓ | x | x | ✓ | x |
| Compcars | x | x | x | ✓ | ✓ |
| Census cars | x | x | x | ✓ | ✓ |
| ImageNet | x | x | ✓ | x | x |
| iNat2017 | ✓ | ✓ | ✓ | x | x |
| Pl@ntNet-300k | ✓ | ✓ | ✓ | ✓ | ✓ |

Table 4: Comparison of several datasets with Pl@ntNet-300k. "Focused domain" indicates whether the dataset is made up of a single category (*i.e.,* cars) and "Ambiguity preserving sampling" indicates whether in the construction of the dataset, all classes belonging to the same parent in the class hierarchy were kept or not (in our case, the parent corresponds to the genus level).

different network architectures [23, 24]. However, these approaches focus on optimizing top-1 accuracy. Set-valued classification, on the other hand, consists in returning more than a single class to reduce the error rate, with a constraint on the number of classes returned. Therefore, FGVC and set-valued classification methods are not mutually exclusive but rather complementary.

Several FGVC datasets, which exhibit visually similar classes, have been made publicly available by the community. They cover a variety of domains: [4], cars [5, 25], birds [26], flowers [3]. However, most of these datasets focus exclusively on proposing visually similar classes (aleatoric uncertainty) with a limited amount of epistemic uncertainty. This is the case for balanced datasets which have approximately the same number of images per class, or with small intra-class variability such as aircraft and cars datasets, where most examples within a class are nearly the same except for angle, lightning, etc... ImageNet [6] has several visually similar classes, organized in groups : it contains many bird species and dog breeds. However, these groups of classes are very different: dogs, vehicles, electronic devices, etc. Besides, ImageNet does not exhibit a strong class imbalance. Several of these datasets were constructed by web-scraping, which can be prone to noisy labels and low quality images. Most similar to our dataset is the iNat2017 dataset. It contains images from the citizen science website iNaturalist. The images, posted by naturalists, are validated by multiple citizen scientists. The iNat2017 dataset contains over 5000 classes that are highly unbalanced. However, iNat2017 does not only focus on plants but proposes several other 'super-classes' such as Fungi, Reptilia, Insecta ... Moreover, the authors selected all classes with a number of observations greater than 20, whereas we choose to randomly sample 10% of the genera of the entire Pl@ntNet database and keep all species belonging to these groups with a number of observations greater than 4. We argue that keeping all species of the same genus maximizes aleatoric uncertainty, as species belonging to a genus tend to share visual features. We summarize the properties of the mentionned datasets in Table 4 and Table 5.

|  | Number of images | Number of classes |
|---|---|---|
| CUB200 | 6,033 | 200 |
| Oxford flower dataset | 8,189 | 102 |
| Aircraft dataset | 10,000 | 100 |
| Compcars | 136,727 | 1,687 |
| Census cars | 712,430 | 2,657 |
| ImageNet | 1,331,167 | 1,000 |
| iNat2017 | 857,877 | 5,089 |
| Pl@ntNet-300k | 306,146 | 1,081 |

Table 5: Number of images and number of classes of several datasets

## 6 Data access and additional ressources

The Pl@ntNet-300K images dataset [27] can be downloaded at:

https://doi.org/10.5281/zenodo.4726653

It is organised in three folders named "train", "val" and "test". Each of these folders contains $d = 1,081$ subfolders. We provide the correspondence between the names of the subfolders and the names of the classes. Class names are of the form *Genus_species*, *e.g., Cymbalaria_aequitriloba*.
We also provide a metadata file containing for each image the following information: the species identifier (class), the organ of the plant (flower, leaf, bark, . . . ), the author's name, the licence and the split (*i.e.,* train, validation or test set).
The github repository containing the code to reproduce the experiments of this paper can be found at:

https://github.com/plantnet/PlantNet-300K/.

It will also be used to report potential issues related to the dataset.

## 7 Conclusion

In this paper, we share and discuss a novel plant image dataset, called Pl@ntNet-300k, obtained as a subset of the entire Pl@ntnet database and intended for evaluating set-valued classification methods. Unlike previous datasets, Pl@ntNet-300k is designed so as to preserve the high level of ambiguity across classes of the initial real-world dataset as well as its long tail distribution. To evaluate set-valued predictors on Pl@ntNet-300k, we examine two main different metrics: top-$k$ accuracy (as a baseline) and average-$k$ accuracy which is a more challenging task requiring to predict sets of various size but still equal to $k$ on average. Our baseline result using a ResNet-50 trained with cross-entropy suggests that there is plenty of room for new set-valued prediction methods that would improve the average-k accuracy over top-k. We hope that Pl@ntNet-300K can serve as a reference dataset for this problem, which is why we created it and share it with the scientific community.

## Acknowledgments

This work was partially funded by the ANR CaMeLOt ANR-20-CHIA-0001-01. It has received funding from the European Union's Horizon 2020 research and innovation program under grant agreement No 863463 (Cos4Cloud project).

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
