# OpenReview forum: "Pl@ntNet-300K: a new plant image dataset for the evaluation of set-valued classifiers"
_NeurIPS.cc/2021/Track/Datasets_and_Benchmarks/Round1 — Submitted to NeurIPS 2021 Datasets and Benchmarks Track (Round 1)_

### Official Review · Reviewer_ADhr · 2021-06-30
**While there are no new technical insights, this work represents a good initial effort to help the research community to evaluate classification methods in real-world ambiguity, I rate it as weakly accept.**

**Rating:** 6
**Confidence:** 4
**Clarity:** Writing quality is good.

**Strengths:**

- First effort at creating a comprehensive dataset with high intrinsic ambiguity and long tailed distribution. Will accelerate research.

- Clear explanations and illustrations.

- In general the paper is easy to follow.





**Weaknesses:**

- Experiments are insufficient to validate the contributions.




**Additional Feedback:**

- It is better to show results based on other classifiers along with resnet50, such DenseNet, MobileNet.
- Make sure the images in the dataset are not recompressed with a lossy-method.

**Correctness:**

This paper described the construction of the dataset, image selection and train/test/val set division is in reasonable way.


**Documentation:**

In the supplementary material, the authors give sufficient detail about collection , organnization and maintennace , etc., and provide a URL to download the dataset. The dataset  will be distributed under Creative-Common Attribution-ShareAlike 2.0 license.

**Ethics:**

There are no ethical concerns for the submission.

**Relation To Prior Work:**

Yes, this paper compares the proposed dateset with previous contributions.

**Summary And Contributions:**

This paper presents a plant image dataset, called Pl@ntNet-300k, and provides detail baseline evaluation based on resnet50. Pl@ntNet-300k is much more comprehensive than previous datasets, which is designed to remain representative of real-world ambiguity ​across classes, combined with the long tailed distribution, makes it more challenging to set-valued classifiers.

---

> ### Author Response · Authors · 2021-07-09
> **We will evaluate other classifiers + information on image compression**
>
> Dear Reviewer,
>
> Thank you for your time and helpful comments.
> We will be using other classifiers as you rightly point out and update the paper accordingly before the discussion phase closes.
>
> Regarding your comment about recompression, we would like to bring some additional information to your attention. The Pl@ntNet application currently represents more than 475 million images. For reasons of storage capacity as well as network bandwidth, the images taken by users are stored in jpeg format. Another motivation is to have a homogeneous quality of the images, both for training and for inference. We hope to have answered your question. If you need further information, we will be happy to provide it.
>
> Best regards

---

> > ### Author Response · Authors · 2021-07-14
> > **We have added other classifiers**
> >
> > Dear reviewer,
> >
> > Thank you again for your time and helpful comments.
> > We have provided results with other classifiers as you suggested.
> >
> > Best Regards

---

### Official Review · Reviewer_51GD · 2021-07-04
**Good paper**

**Rating:** 6
**Confidence:** 3
**Correctness:** Yes.
**Clarity:** Yes, the paper is well written.

**Strengths:**

The proposed dataset has a strong class imbalance, meaning that a few species account for most of the images. And many species are visually similar, making identification difficult even for the expert eye. These two characteristics make the present dataset well suited for the evaluation of set-valued classification methods and algorithms.

**Weaknesses:**

The author only used a resnet50 as baseline. I wonder it is not the SOTA model in set-valued classification.

**Additional Feedback:**

 It would be better if there are more results about SOTA methods performance on the proposed dataset.

**Documentation:**

Yes. But more checks maybe needed.

**Ethics:**

No.

**Relation To Prior Work:**

Yes, but it would be better if there are more discussions and comparation between set-valued classification and Fined-Grained Visual Classification (FGVC).

**Summary And Contributions:**

This paper presents a novel image dataset with high intrinsic ambiguity specifically built for evaluating and comparing set-valued classifiers.  The author recommend two set-valued evaluation metrics associated with the dataset (top-k and average-k) and provided the results of a baseline approach based on a resnet50 trained with the cross-entropy loss.

---

> ### Author Response · Authors · 2021-07-09
> **We will evaluate other backbones and develop the comparison between Set-valued classification and FGVC**
>
> Dear Reviewer,
>
> Thank you for your time and helpful suggestions.
> We will evaluate other models as you rightly point out. We appreciate your comment on the comparison between set-valued classification and FGVC, as it gives us the opportunity to provide additional insight to the reader. We will explain this in the « Related work » section before the discussion phase closes.
>
> Regarding the comment on the documentation, we would be happy to add any information you feel would be helpful in the supplementary material.
>
> Best Regards

---

> > ### Author Response · Authors · 2021-07-14
> > **We have added several architectures and the comparison between FGVC and set-valued classification**
> >
> > Dear Reviewer,
> >
> > Thank you again for your time and helpful suggestions.
> > We have added several model evaluations to the paper as you suggested. We have also detailed the distinction between set-valued classification and FGVC in the "Related Work" section.
> >
> > Best regards

---

### Official Review · Reviewer_rKAj · 2021-07-05
**Pl@ntNet-300K: a new plant image dataset for the evaluation of set-valued classifiers**

**Rating:** 6
**Confidence:** 3
**Correctness:** The paper is easy to follow.
**Clarity:** The paper is easy to follow.

**Strengths:**

The paper proposes a comprehensive dataset with high intrinsic ambiguity and long tailed distribution, and the paper is well written and easy to follow.

**Weaknesses:**

It is better to provide more experiments to validate the contributions and the proposed dataset. Other backbones should also be used.

**Additional Feedback:**

It is better to provide more analysises of previous papers, and more experiments should be provided, such as different backbones.

**Documentation:**

Yes.

**Relation To Prior Work:**

Yes, and it is better to provide more analysises.

**Summary And Contributions:**

This paper presents a dataset, named Pl@ntNet-300k, for plant image dataset, and it also recommend two set-valued evaluation metrics associated with the dataset.

---

> ### Author Response · Authors · 2021-07-09
> **We will evaluate other backbones and develop the "Related work" section**
>
> Dear reviewer,
>
> Thank you for your time and insightful comments.
>
> We will use other backbones as you rightly suggest and update the paper accordingly before the discussion phase closes. We will also expand the "Related Work" section. If there is anything specific you would like to see in this section, we would be delighted to discuss it with you.
>
> Best Regards

---

> > ### Author Response · Authors · 2021-07-14
> > **We have added additional backbones and expanded the "Related Work" section**
> >
> > Dear reviewer,
> >
> > Thank you again for your time and insightful comments.
> > We have conducted additional experiments using different backbones as you suggested.
> > We have also added a table highlighting some properties of the compared datasets.
> >
> > Best regards

---

### Decision · Program_Chairs · 2021-07-26

**Decision:**

Reject

**Comment:**

While the proposed dataset could be quite valuable to the community, there are a few experiments that would significantly strengthen the paper. Currently, it's not clear how difficult/easy the dataset is, what the relationship is between avg-k and top-k, and whether ImageNet models would perform well on the dataset without much modification.

Overall, we recommend the authors resubmit after adding the following experiments:

1. scatter plot of ImageNet versus PlanetNet accuracy for at least 20 models covering decent range of accuracies
2. scatter plot for top-k versus avg-k metric to see if one is just a function of the other
3. some estimate of Bayes risk (e.g. by evaluating a trained human labeler) so that we have a sense of how difficult /easy the dataset is